# Comment on K. Michaelian and A. Simeonov (2015) "Fundamental molecules of life are pigments which arose and co-evolved as a response to the thermodynamic imperative of dissipating the prevailing solar spectrum".

Lars Olof Björn[1]

[1]Department of Biology, Lund University, SE-224 67 Lund, Sweden

*Correspondence to*: Lars Olof Björn (lars_olof.bjorn@biol.lu.se)

**Abstract.** This is a comment on: "Fundamental molecules of life are pigments which arose and co-evolved as a response to the thermodynamic imperative of dissipating the prevailing solar spectrum" by K. Michaelian and A. Simeonov, 10 Biogeosciences, 12, 4913–4937, 2015. Michaelian and Simeonov formulate the leading thought in their article: "The driving force behind the origin and evolution of life has been the thermodynamic imperative of increasing the entropy production of the biosphere through increasing the global solar photon dissipation rate". I shall in the following try to provide some information that might help to clarify whether this is correct.

## 1 Introduction: Do living systems reduce the albedo of Earth?

Already in the first sentence of their abstract Michaelian and Simeonov formulate the leading thought in their article "The driving force behind the origin and evolution of life has been the thermodynamic imperative of increasing the entropy production of the biosphere through increasing the global solar photon dissipation rate".

As long as light travels freely in space, there is no change in the entropy associated with it. If it is scattered, entropy is increased. If we consider only the non-absorbed fraction of the light, a surface reflecting light in a diffuse way (i.e., non-20 specularly), will increase entropy by scattering, not by changing the photon number; there is, in principle, no difference between non-living substances (e.g., pure sand) and organisms in this respect. Thus we can focus on absorption. An absorbing surface on Earth will eventually cause incident radiation to be converted to diffuse radiation of an increased number of less energetic photons, and thus increase entropy more than a reflecting surface (see, e.g., Delgado-Bonal, 2017), although processes such as photosynthesis can cause a considerable time-delay in entropy production. Thus, it appears that if Michaelian 25 and Simeonov are correct, one would expect organisms, in particular phototrophic organisms, or the biosphere), to be less reflecting and more absorbing than dead matter. They explicitly state: "Living systems reduce the albedo of Earth". Is this correct?

## 2. Ancient life

Methanogens represent one of the earliest life-forms on our planet, although not the earliest one (Muñoz-Velasco, 2019), and Ueno et al., 2006, estimate that they existed at least 3.46 Ga (gigayears) ago. Their emission of methane to the atmosphere resulted, under the influence of sunlight, in a colored haze of organic compounds, somewhat similar to that now seen on Titan, the largest moon of Saturn (e.g., Arney et al., 2016). It is likely that a thick haze existed continuously from 3.2 to 2.7 Ga ago (Domagal-Goldman et al., 2008). Later, "on the eve of the great oxidation event", 2.7–2.5 Ga ago, the atmosphere oscillated

between a hazy state and periods without thick organic haze (Izon et al., 2015). During the hazy periods no ultraviolet radiation, and only a small fraction of incident visible light would have penetrated to ground and ocean levels (Arney et al., 2017).

The geometric albedo of an astronomical body is the ratio of its total brightness, looking along the direction of illumination, to that of an idealised fully reflecting, diffusively scattering (Lambertian) disk with the same cross-section. The geometric spectral albedo of the resulting haze depends on the $CH_4/CO_2$ ratio: The higher it is, the higher is the peak wavelength

for the reflectance, and the higher the albedo maximum (Arney et al., 2016). For a $CO_2$ pressure of 0.018 bar, at total pressure of 1 bar, and a $CH_4/CO_2$ ratio of 0.21, the haze has a broad reflectance band with a maximum albedo of 0.22 at 750 nm. These values are very sensitive to the $CH_4/CO_2$ ratio. Pavlov et al. (2001) assume a $CH_4/CO_2$ ratio around 1, which can be assumed to have resulted in a thicker haze with a higher reflectance peak at higher wavelength.

Various other authors date the oldest signs of life as being 3.22 Ga (Homann, 2019), 3.5 (Djokic et al., 2017, Lepot,

2020), 3.80 Ga (Mojzsis et al., 1996), 3.77 or possibly 4.28 Ga old (Dodd et al., 2017). Hoashi et al. (2009) conclude: "These data strongly suggest that oxygenic photoautotrophs flourished in the photic zone of the 3.46 Gyr oceans and supplied molecular oxygen to the deep water." So it is quite possible that other organisms preceded the methanogens. In any case, it is very unlikely that life could have survived prior to 4.4 Ga ago, because of the formation of the Moon and the "late heavy bombardment" (Bottke and Norman, 2017).

The estimate of the oldest life above, that of Dodd et al., 2017, as well as that of Djokic et all, 2017, concern life in hydrothermal vent precipitates. There are many other publications suggesting an origin of life in connection with hot springs (see Omran and Pasek, 2020), often at the bottom of the sea (e.g., Martin et al., 2008), out of reach for sunlight. But photosynthesis, and even the complicated oxygenic version, originated very early (see above). These views were recently discussed by Damer and Deamer, 2020.

## 3 Present vegetation compared to bare ground

We must not be restricted to visible light, but extend our interest into the infrared, a spectral region to which much of the solar radiation belongs. The quantity we should consider as far as data are available is the hemispherical reflectance, not reflectance in a single direction. Unfortunately, in most cases values of hemispherical reflectance are not available. Directional reflectance spectra (Figure 1) seem to indicate that vegetated areas may reflect more light throughout the daylight spectrum. The kind of

terrestrial life most similar to the newcomers on land may be "biocrusts". As shown by the example in Figure 2, these crusts

can have albedos exceeding that of the bare ground they inhabit, with an exception only for a very dry state, with less biological activity. There are also cases in which even simple organisms decrease the albedo, one reason being the production of pigments protecting them from sunlight (e.g., Couradeau et al., 2016), a process for which the solar ultraviolet radiation is of particular importance. This fact does not prove that pigments are formed due to a "thermodynamic imperative" for increasing entropy.

The presence of moss strongly decreases soil surface albedo, while lichens often have the opposite effect (Blanco-Sacristán, 2019; Xiao and Bowker, 2020)

We shall not go into details of the complications associated with evaporation and other phase transitions; just point out that forests have a tendency to increase cloudiness, and thereby cause increased reflectance by the atmosphere (e.g., Teuling et al., 2017), and thereby counteract entropy increase (while the evaporation itself is associated with an entropy increase).

**4 The temporal aspect**

We can also put a temporal aspect on this. If sunlight is absorbed by dead matter, it is usually converted to heat and reradiated as heat radiation of ambient temperature within a short time. If it is absorbed by a photosynthetic system, much of the energy is retained for a long time, before eventually being radiated as heat, sometimes after having been processed through several steps in the food chain. Thus, the living system delays entropy increase. Some trees that had collected solar energy sank into

swamps more than two hundred fifty million years ago, and their reduced carbon was preserved until the present era, when mankind started to burn coal and thereby generate entropy. Thus, entropy production was delayed, i.e. the rate of production

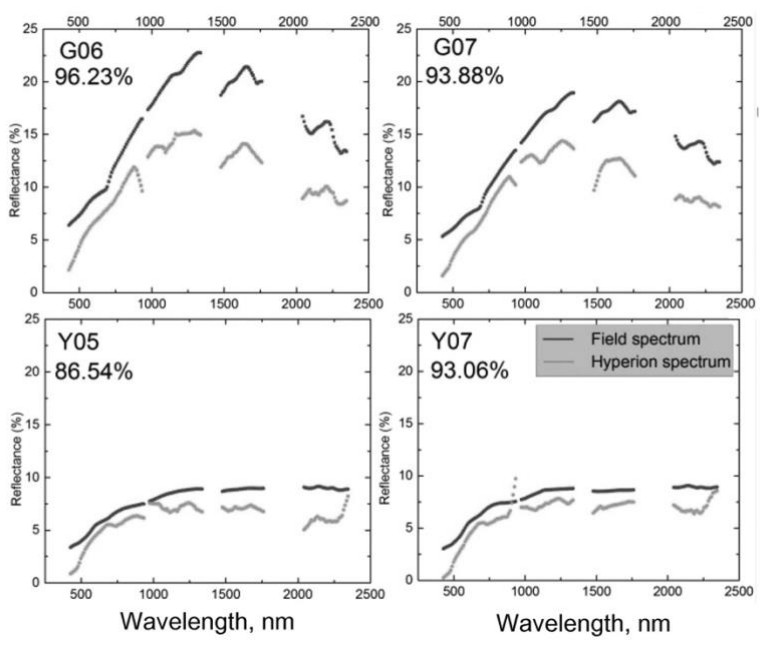

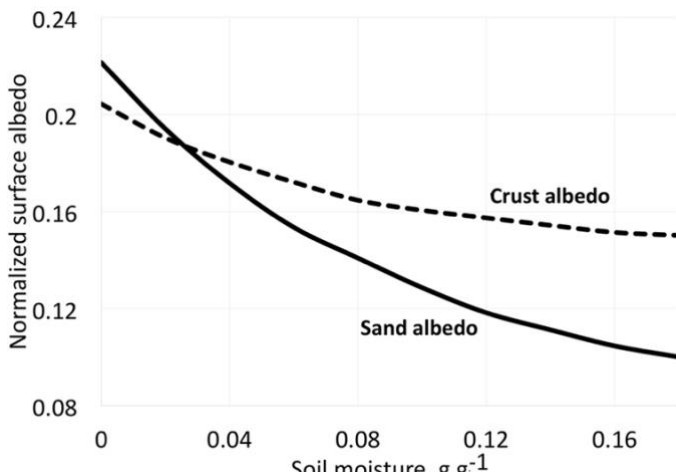

Figure 1. Reflectance spectra from the volcano Teide on Tenerife, one of the Canary Islands. The spectra were recorded on the ground (black) or remotely from satellite (grey). The percent values indicate the agreement between the red and green curves. Top left (G06) is from a site with heavy lichen cover but no plants, top right (G07) heavy lichen cover and some plants. Bottom panels show spectra of bare lava (no vegetation). From Li et al. (2015). Creative Commons Attribution License (http://creativecommons.org/licenses/by/4.0/).

Figure 2. Albedo for desert dune sand with and without biological soil crust organisms (BSC) at different moisture contents. Redrawn and simplified from Zhang et al. (2013).

decreased, thanks to the chlorophyll and the photosynthesis of ancient trees that have been preserved as coal. The production and usage of oil is a similar phenomenon.

## 5 The aquatic environment

Much of the Earth surface is covered by water, life probably originated in water, and much of life's evolution has taken place in water. Thus we must also consider the reflectance of water bodies, and how their reflectance is affected by life. In addition, aquatic life is easier to deal with, since we do not have to deal with gases (with the exception of dissolved ones, which may escape to the atmosphere, and those forming froth), and processes such as transpiration. Bach et al. (2021) suspect that the increased albedo caused by recent increase in the Atlantic Ocean of floating *Sargassum* algae may be more important than the effect of the alga on carbon dioxide and phytoplankton nutrients. A minor effect might arise from increased temperature in the presence of organisms at the surface, and probably some increased evaporation (Kahru et al., 1993). However, one should also take into account that the heating that does not take place at the surface if algae are absent there, takes place deeper down, in a more diffuse way.

Aquatic organisms do not decrease the reflectance of the ocean and other water bodies. This is not always as clear as in the data of Qi et al. (2020), who have published numerous reflection difference spectra for various lakes and ocean surfaces where algae are abundant, i.e. spectra that show the difference in reflectivity for water with and without algae. A sample is

reproduced in Figure 3. In many other cases algae cause a decrease of reflectance at short wavelengths, and an increase above
about 500 nm, both for phytoplankton (Jin et al., 2002, 2011;  Bacour et al., 2020,; Li et al., 2021) (Figure 5) and for brown

105     and green macroalgae (Qi and Hu, 2021). However, taking the daylight spectrum into account, one finds that the algae increase
the total light reflection. The data in Figure 4 do not extend over a sufficiently wide wavelength range to permit such a

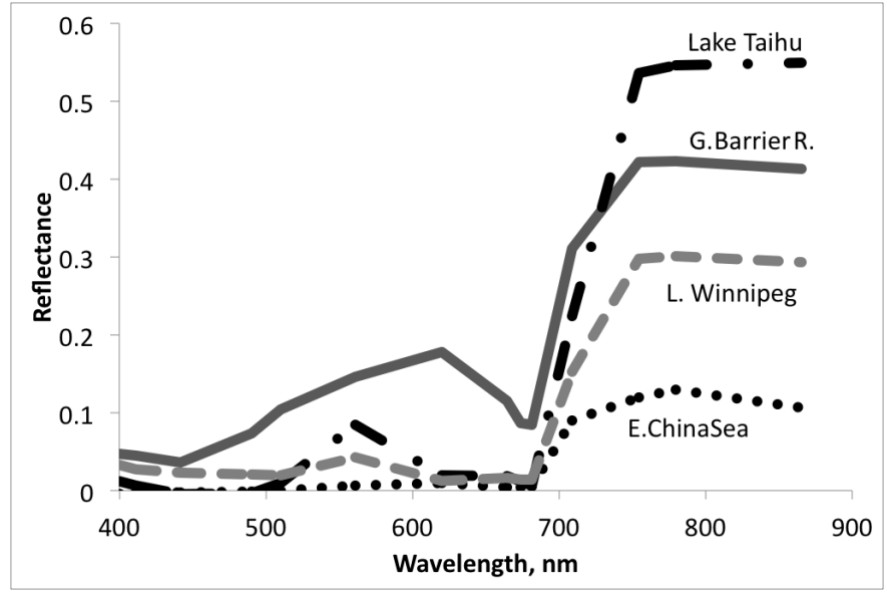

**Figure 3. A sample of sea and lake reflectance difference spectra redrawn from Qi et al. (2020). The diagram shows reflectance of waters with algae minus adjacent water without algae. These differences are mostly positive, in some cases with the exception for**

110     **very small negative excursions at short wavelengths.**

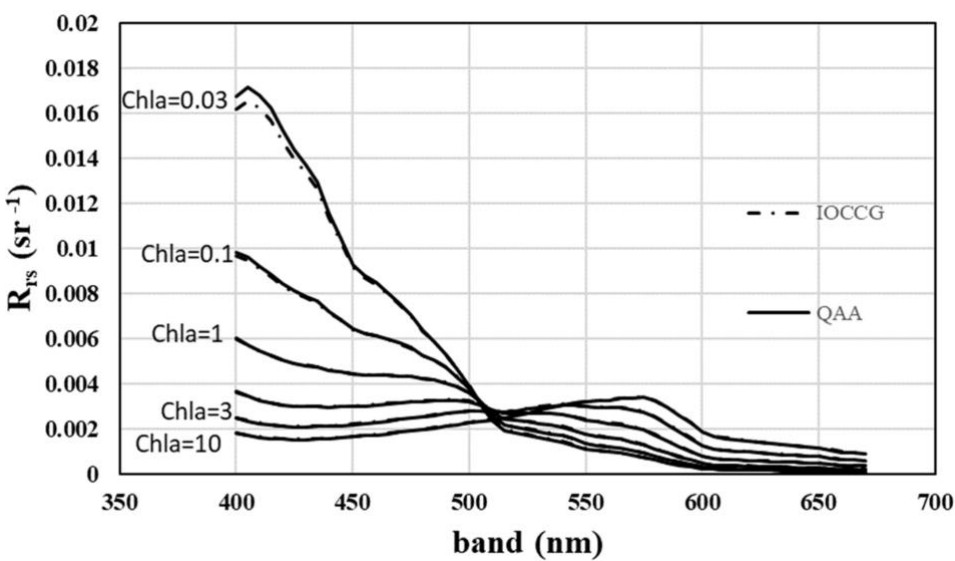

**Figure 4. Ocean reflectance over 1 radian according to Li et al. (2021) for various concentrations of chlorophyll *a*-containing organisms using two different models. Chlorophyll *a* concentrations in mg m⁻³. Creative Commons Attribution (CC BY) license (https://creativecommons.org/licenses/by/4.0/).**

comparison, but we show the calculation for Bacour et al. (2020) and Jin et al. (2002) in Figure 5. Both based on the data of Bacour et al. (2020) and those of Jin et al. (2002) more light is reflected from water containing more algae. This holds on an energy basis, and the increase in reflected light by the algae is even more pronounced when expressed on a photon basis. The light that is not reflected (or emitted as fluorescence) is eventually converted to heat radiation.

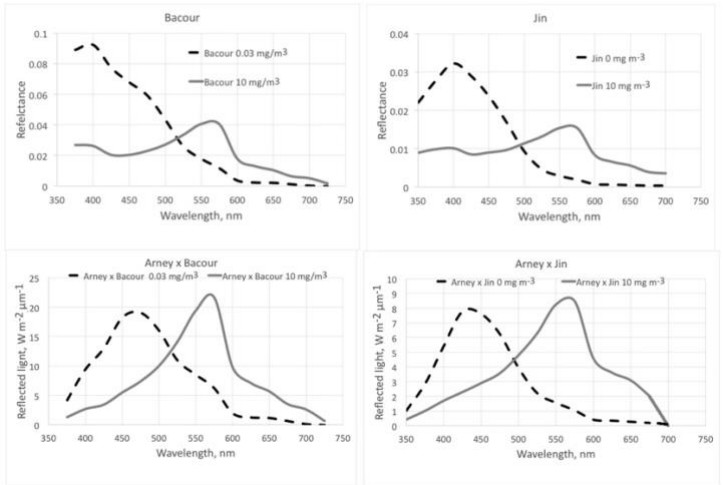

**Figure 5. The effect of phytoplankton on reflectance of the ocean surface. The dashed lines are for a low chlorophyll content of the surface layer (0 or 0.03 mg m⁻³), the solid lines for a high chlorophyll content (10 mg m⁻³). The top two graphs show the modeled spectra from Bacour et al. (2020) and Jin et al. (2011, diffuse incidence). The geometry is not the same for the two panels, thus the absolute values are not directly comparable. The lower two graphs show the amount of reflected light, assuming an incident Archaean daylight according to the red graph in Fig. 15 (right panel) of Arney et al. (2016).**

Thus the presence of algae increases the reflectivity of oceans and lakes, i.e. counteracts the "degradation" of sunlight to diffuse radiation of longer wavelength in all directions (not only from the sunlit side of the planet), and the production of entropy.

Aquatic organisms also have indirect effects on the optical properties of both water surfaces and atmosphere. Substances released by algae, (e.g., *Phaeocystis globosa*, Blauw et al. 2010), bacteria (e.g., Rahlff et al. 2021), or other organisms may cause formation of light-reflecting foam on the surface. A much more important effect is the biological increase of cloudiness. In addition to the effect on cloudiness by forest transpiration already mentioned, there is a biological effect on condensation nuclei which results in increased aerosol and increased cloudiness. Algae form dimethylsulphoniopropionate (DMSP) which decomposes to dimethylsulphide (DMS). In the atmosphere this is converted to various compounds, of which sulfur trioxide and sulfuric acid in particular contribute to aerosol formation and increased reflection of incoming sunlight, thereby decreasing entropy production. I cannot agree with Michaelian & Simeneov (2015) that always: "Living systems reduce the albedo of

Earth and dissipate, through many coupled irreversible processes, shortwave incoming radiation into long-wave radiation, which is eventually returned to space, ensuring an approximate energy balance in the biosphere".

## 6 Ice and snow

On the other hand algae on the snow decrease the albedo, in one investigation (Lutz et al., 2016) from 0.7 down to, in one case, as low as a 0.5. Yallop et al. (2012) report that under conditions when algae, in addition to photosynthetic pigments, form protective pigments, they decrease Greenland ice albedo from 0.60 (clean ice) to 0.26.

## 6 Conclusion

Michaelian & Simeneov (2015) conclude that they have presented evidence that "supports the thermodynamic dissipation theory of the origin of life (Michaelian, 2009, 2011), which states that life arose and proliferated *to carry out* the thermodynamic function of dissipating the entropically most important part of the solar spectrum (the shortest wavelength photons) prevailing at Earth's surface and that this irreversible process began to evolve and couple with other irreversible abiotic processes, such as the water cycle, to become more efficient, to cover ever more completely the electromagnetic spectrum, and to cover ever more of Earth's surface." I cannot agree that they have presented evidence for this conclusion. The biosphere has certainly evolved and is maintained thanks to a production of entropy associated with the conversion of solar radiation to Earth radiation. For details of this entropy production, I refer to Wu and Liu (2010). They compared and discussed various ways of computing entropy fluxes in the Earth system. There are many examples of how organisms contribute to increased albedo and decreased entropy production.

## 7 Final comments

Ultraviolet radiation has played an important role in the prebiotic synthesis of the molecular species that have made the origin of life possible (Björn 2015). But several recent publications explore the possibility that the first life emerged in places without any ultraviolet radiation or visible light, such as submarine hydrothermal vents (Altair et al., 2021) or a geyser system driven by nuclear power (Maruyama et al., 2019).

## 8 There are no competing interests

## 9 Acknowledgements

Thanks are due to Beth Middleton for comments and language correction, and to an anynomous reviewer for helpful comments.

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
