# Peer review of "Comment on K. Michaelian and A. Simeonov (2015) “Fundamental molecules of life are pigments which arose and co-evolved as a response to the thermodynamic imperative of dissipating the prevailing solar spectrum”."

_Biogeosciences, 2021_

## Community Comment (CC4)

**Reply to Lars Olof Björn's Comment on our article "Fundamental molecules of life are pigments which arose and co-evolved as a response to the thermodynamic imperative of dissipating the prevailing solar spectrum"**

Karo Michaelian[1] and Aleksandar Simeonov[2]

[1]Department of Nuclear Physics and Application of Radiation, Instituto de Física, Universidad Nacional Autónoma de México, Circuito Interior de la Investigación Científica, Cuidad Universitaria, Cuidad de México, C.P. 04510
[2]Faculty of Natural Sciences and Mathematics, Ss. Cyril and Methodius University in Skopje, North Macedonia

**Correspondence:** Karo Michaelian (karo@fisica.unam.mx) and Aleksandar Simeonov (alecsime.gm@gmail.com)

**Abstract.** Lars Björn, in his critique of our article, doubts our assertion that at the origin of life the fundamental molecules of life (those in the three domains) were UVC pigments, dissipatively structured under a thermodynamic imperative to absorb and dissipate this UVC light into heat. Björn bases his critique on the suggestion that non-living material can be more photon absorbing than living material. He gives a number of examples in which he shows that the albedo of material devoid of life is lower than that of biotic material and concludes that these examples counter our assertion. However, Björn makes the erroneous assumption that albedo (reflection) is the only important factor related to photon dissipation (entropy production) occurring in the light-pigment interaction in living systems. He ignores the other contributions to entropy production due to the photon interaction which were listed in our article; 1) the shift towards the infrared of the emitted spectrum, 2) the diffuse emission and reflection of light into a greater outgoing solid angle, 3) the coupling of photon-induced evapotranspiration in the pigmented leaf to further photon dissipating processes such as the water cycle, which further allows dissipating biopigments to flourish over all of Earth's surface. His assumption is therefore incorrect and his analysis does not provide legitimate reason for doubting our assertion that the fundamental molecules of life arose as pigments as a response to the thermodynamic imperative of dissipating the prevailing solar spectrum.

In the following, we respond to each critique using the same section headings of Björn's Comment.

**1 Introduction: Do living systems reduce the albedo of Earth?**

Contrary to Björn's assertions, living organisms do, in fact, generally reduce the albedo with respect to regions devoid of life, and there are many data available in the literature. For example, the visible albedo of deciduous forests is 0.15 to 0.18 and that of coniferous forests is 0.09 to 0.15, while that of sandy deserts is about 0.30 [Barry and Chorley (1992)] and rocky deserts (Gobi) 0.21 [Wang et al. (1998)]. And this is also true at wavelengths greater than the red-edge ($\sim 700$ nm), for example, forest

albedo increases to about 0.3 [Coakley (2003)], while sand and rocky desert albedo increases to about 0.50 [Varotsos et al. (2014); Coakley (2003)].

Our main objection to Björn's critique of our paper, however, is that it is based on his erroneous assumption that albedo is the only important factor relevant to photon dissipation in the light-pigment interaction. Björn states, "Thus, it appears that if Michaelian and Simeonov are correct, one would expect organisms, in particular phototrophic organisms, or the biosphere to be less reflecting and more absorbing than dead matter." Although, as stated above, this is generally the case, one should not *a priori* "expect" this since, as we mentioned even in the abstract of our original article [Michaelian and Simeonov (2015)], albedo (reflected light) is only one part of the equation for determining global photon dissipation or entropy production which can be attributed to pigments in life. Other factors important to entropy production are; (1) the red-shifting of the absorbed energy in the pigments, (2) dispersion of the emitted, reflected, and transmitted photon beams into a larger outgoing solid angle, and (3) the coupling of photon dissipation in pigments to other abiotic entropy producing processes, such as the water cycle [Michaelian (2012a, b)] which, i) further red-shifts the energy, and ii) allows greater proliferation and spread of organic pigments over an ever greater surface area of Earth.

Therefore, the plausibility of our assertion that the fundamental molecules of life arose as pigments and co-evolved in response to the thermodynamic imperative of dissipating the prevailing solar spectrum could only be evaluated by considering all the factors related to photon dissipation (entropy production) due to pigments, not only the albedo. This can easily be achieved by comparing the incoming solar spectrum with the outgoing global Earth reflected and emitted spectra, determining the entropy production from the differences of these spectra, and then comparing this to other similar bodies devoid of life. We have, in fact, performed such detailed calculations of the entropy production of the Earth and compared it to that of its sister planets Venus and Mars in another paper [Michaelian (2012b)] cited in our article under discussion. We found that Earth's entropy production per unit area is almost twice that of either of its neighbors, and this may be attributed to the presence of life on Earth [Michaelian (2012b)]. Kleidon et al. (2000) have compared the surface temperatures and amount of water vapor of a simulated Earth with and without life and find 8 °C average lower temperature and 3 times the amount of water vapor in the atmosphere for the simulation with life; the lower temperature and greater amount of water vapor implying (see below) significantly greater entropy production for a planet with life.

**2   Albedo of the Moon, a world without life**

Comparing the albedo of Earth to the Moon, in the manner Björn does to demonstrate that non-living material is better at dissipating photons than living material, is not valid because, 1) the Earth and moon are physically very different astronomical bodies, and, more importantly, 2) albedo is a poor proxy for photon dissipation and entropy production. We can see this by performing a very simple approximate black-body calculation (see Michaelian (2012b) for a more accurate grey-body calculation) which takes into account, not only the albedo (reflected light), but also the emitted light due to the red-shift in the pigments, emission into greater solid angle, and the additional red-shift due to photon dissipation in pigments coupling to abiotic processes such as the water cycle.

The temperature that the Earth emits into space, including the effect of the coupling of the heat of photon dissipation in pigments to the water cycle (evapotranspiration) and other irreversible processes (winds, ocean currents, convection cells, etc. ), is approximately -18 °C (255 K) corresponding to the temperature of the middle of the troposphere, at the cloud tops ($\sim$ 5 Km), averaged over latitude. Since average day and night temperatures of the middle troposphere are similar (varying by < 0.5 K [Muhsin et al. (2017)]) due to surface convection, winds, currents, the heat capacity of water vapor, etc., the radiation re-emitted from Earth is effectively emitted into a $4\pi$ solid angle.

The surface temperature of the Moon during daylight hours (days are $\sim$ 30 Earth days) reaches 127 °C (400 K) which is therefore the approximate black-body temperature at which the Moon surface will radiate (re-emit) into space. Since the temperature difference between day and night on the Moon is so large (127 °C to -183 °C) and since the total amount of emitted radiation goes as $T^4$ for black bodies, the majority (99.8 %) of the radiation emitted by the Moon into space occurs during the day and thus radiated basically only into a $2\pi$ solid angle.

Not all light is absorbed by the respective surface, some of it is reflected (albedo). To simplify the calculation, we assume that the albedo is the same over all wavelengths (a wavelength dependent albedo will not change the result). Therefore, we assume that the Moon reflects 15% of all incident light and the Earth 29%. We assume, furthermore, that for both bodies this reflected light is diffuse and reflected into a $2\pi$ solid angle.

Since [Prigogine (1967)],

$$dS = dE \left( \frac{1}{T_1} - \frac{1}{T_2} \right),\tag{1}$$

the approximate ratio of the entropy production Earth/Moon per unit incident energy per unit surface area for emission plus reflection is,

$$\frac{dS_{Earth}}{dS_{Moon}} \approx \frac{0.71 \cdot \frac{4\pi}{\Omega_{Sun}} \cdot \frac{1}{(273-18)} + 0.29 \cdot \frac{2\pi}{\Omega_{Sun}} \cdot \frac{1}{5800}}{0.85 \cdot \frac{2\pi}{\Omega_{Sun}} \cdot \frac{1}{(273+127)} + 0.15 \cdot \frac{2\pi}{\Omega_{Sun}} \cdot \frac{1}{5800}} = \frac{0.0349887 + 3.1415926 \times 10^{-4}}{0.0133517 + 1.6249617 \times 10^{-4}} = \frac{0.0353028}{0.0135142} = 2.612 \tag{2}$$

where we are using a Sun surface temperature of 5800 K and $\Omega_{Sun}$ is the solid angle subtended by the Sun on Earth (or Moon), $7.193 \times 10^{-5}$ sr, but we have set it to one for the numerical values given in equation (2) since it anyway drops out of the ratio. We have also ignored the term $-1/T_2 = -1/5800$ in the first terms on the right hand side of the equation.

The entropy in the emitted plus reflected light per unit incident energy, per unit surface area, per unit time, is thus about 2.6 times greater for the Earth than for the Moon, contrary to what Björn would conclude from his albedo comparison. This is because the black-body spectrum emitted at a lower temperature by Earth is much more red-shifted than that emitted by the Moon at a much higher temperature, and also because the Earth emits its absorbed solar energy into a $4\pi$ instead of $2\pi$ solid angle (in large part due to life increasing the amount of water vapor in the atmosphere [Kleidon (2008)]). Thus, even though the Moon absorbs more light (has lower albedo), this increase in absorption does not compensate for the greater amount of red-shift and the dispersion of the emitted light into a greater solid angle occurring on Earth attributable to pigments in life. Simply comparing albedos of different materials, therefore, says very little about their photon dissipation potential, especially if one of them is living material.

**3  Vegetation compared to bare ground**

Björn's second comparison of vegetation with "bare ground" suffers from the same oversight since it again considers only albedo (reflection). Rather than looking only at reflection data, it is also important to consider the temperature data of the emitted spectrum. There exists extensive data for this [Schneider and Kay (1994)] compiled from infrared temperature measurements obtained from airplane fly-overs. The result is clear; the temperature measurements over climax ecosystems are lower than those measured over perturbed ecosystems, and these are lower than those measured over regions devoid of life. Another simple way of seeing this is that rocks (or ground without organic material) become much hotter under the sun than does vegetation – albedo plays only a small part, it is the association of life with water and the water cycle that plays the greater part.

Even beyond the red-edge, however, as mentioned above, the albedo of areas covered with vegetation is usually lower than that devoid of life [Barry and Chorley (1992); Wang et al. (1998); Varotsos et al. (2014); Coakley (2003)].

A further note of caution is that "bare ground" is usually not devoid of life, or life produced (biological) pigments, and water. Sufficiently developed biocrusts reduce significantly the albedo of the soils they cover [Ustin et al. (2009)]. An important component is the cyanobacterial pigment scytonemin which reduces albedo significantly [Couradeau et al. (2016)].

**4  The temporal aspect**

The forests, as Björn correctly indicates, are sometimes buried and later burned as fossil fuel by humans. However, they produced at least 1000 times more entropy during their lifetime than that obtained by burning the same trees as fossil fuel today. Less than 0.1% of the free energy in sunlight goes into carbon bond making, which is how photosynthesis stores free energy [Gates (1980)]. In living plants, more than 99.9% of solar free energy is simply turned into heat through photon dissipation in the leaves (the dissipation involved in the process of photosynthesis itself, plus non-photochemical quenching). This heat of dissipation is then coupled by the living system to the water cycle through evapotransporation from leaves which increases further the photon dissipation or entropy production of Earth [Michaelian (2012b)].

The fact that a small amount of free energy available in sunlight is not instantly dissipated by ecosystems, and instead is stored for different amounts of time, has no bearing on the point under discussion concerning whether or not pigments, life, and ecosystems arose as a result of the thermodynamic imperative of entropy production through photon dissipation. Storage of free energy for later use is, of course, necessary for maintaining the different trophic levels of an ecosystem, and this hierarchy actually improves global dissipation.

Although the storage of a small amount of free energy for larger times (for example, as coal and petroleum) may make ecosystems imperfect at dissipation, so too does; 1) the fact that their albedo is not zero, 2) that pigments absorb less strongly at wavelengths greater than the red-edge, 3) that the physical size of the pigments are not at their theoretical limit, 4) that fluorescence reduces entropy production, i.e. the quantum efficiency for dexcitation through a conical intersection to the ground state could be further increased, 5) that pigment distribution over the whole Earth surface could be further improved. In other words, ecosystems still have room to evolve under the thermodynamic imperative towards becoming even better dissipating

systems. Modern ecosystems are, however, much more apt at dissipating sunlight than were ancient ecosystems, and this can be seen, for example, in the appearance of new pigments over time, in the spread of life over the whole Earth surface, in the fact that increases in vegetation increase water vapor in the atmosphere and this maintains day and night temperatures similar, thereby increasing the solid angle of the Earth's emitted radiation, and, in the fact that much less free energy stored in carbon bonding in organic matter is being buried today as compared to ecosystems of the past.

Nature's thermodynamic imperative of increasing global entropy production through increasing photon dissipation is indifferent to, but at the same time probably the source of, human concerns over responsibility or irresponsibility for burning fossil fuels or for preserving present ecosystems. Although most of us are blissfully unaware of it, the second law is also driving human evolution and activity. Human free energy use (dissipation) has increased exponentially over the last centuries and this trend will continue for as long as we remain a robust knowledge possessing and technical species on this planet. Our future contribution to global dissipation would appear to go much beyond our dissipation of the chemical potential stored in fossil fuels, and beyond our traditional role as gardeners for the plants, to spreading biopigments over the whole of Earth (global greening), and even to eventually terraforming other planets.

**5 Aquatic environments**

Contrary to what Björn suggests, living organisms and free-floating, biologically-derived organic pigments – colored dissolved organic matter, CDOM – at the ocean surface microlayer certainly do augment photon dissipation (entropy production) compared to water without organic material by, 1) increasing photon absorption at the surface, particularly for shorter wavelengths and at shallow incident photon angles, and, 2) increasing the red-shifting of the absorbed energy by coupling it to evaporation from the ocean surface microlayer (see Michaelian (2012b) and references therein). A detailed calculation of the entropy production as a function of incident photon wavelength for the ocean surface microlayer, with and without organic material, is given section 6 of Michaelian (2012b). By absorbing and dissipating UV and visible light, the organic matter at the sea surface microlayer contributes an additional approximately 23% to the total entropy production due to photon dissipation in this layer on a clear day, and on an overcast day, it contributes an additional surprising 400% [Michaelian (2012b)]. The particular data presented by Björn on visible reflection alone, again, says little about photon dissipation or entropy production.

Forests do indeed have a tendency to increase cloudiness over land, but this is important for the other important entropy producing process of the biosphere; the water cycle. Furthermore, cloudiness, although diffusely reflecting sunlight (also producing entropy), increases the probability of rain further inland from the coasts and thus the possibility of sustaining vegetation for photon dissipation far inland from ocean shores [Makarieva and Gorshkov (2007)]. Vegetation, by increasing the amount of water vapor in the atmosphere [Kleidon et al. (2000)], also helps keep the temperature similar on the day and night sides of Earth, meaning emission of infrared radiation into a solid angle of $4\pi$ rather than of $2\pi$, again increasing entropy production.

A grey-body calculation of the entropy production of the Earth compared to its neighbors Mars and Venus shows that Earth's entropy production per unit surface area is almost twice that of Mars (even though, as Björn correctly points out, Mars has

lower global albedo than Earth), and about 1.6 times that of Venus [Michaelian (2012b)], and this is most likely attributable to the presence of life on Earth.

**6 Conclusions**

Björn incorrectly assumes that albedo (reflection) is the only important factor related to photon dissipation in pigments giving rise to entropy production of living organisms, ecosystems, and the biosphere. He ignores the other components involved in entropy production attributable to photon interaction with biology mentioned in our original article and abstract; the shift towards the infrared of the emitted spectrum, the emission into a greater solid angle, covering all of Earth's surface with pigments, the coupling of life to other photon dissipating processes such as the water cycle. His conclusions are therefore mistaken and do not provide legitimate reason for doubting our assertion that "we have presented evidence that supports the thermodynamic dissipation theory of the origin of life [Michaelian (2009, 2011, 2016, 2017, 2021)], which states that life arose and proliferated to carry out the thermodynamic function of dissipating the entropically most important part of the solar spectrum (the shortest wavelength photons) prevailing at Earth's surface and that this irreversible process began to evolve and couple with other irreversible abiotic processes, such as the water cycle, to become more efficient, to cover ever more completely the electromagnetic spectrum, and to cover ever more of Earth's surface."

Finally, since our first articles, published beginning in 2005 [Michaelian (2005, 2009, 2011)], we have continued to uncover evidence pointing to a connection between photon dissipation and the origin and evolution of life. These include; 1) that many of the fundamental molecules of life strongly absorb UVC light in exactly that wavelength region that was arriving at Earth's surface during the Archean [Michaelian (2012b, a); Michaelian and Simeonov (2015); Michaelian (2016)], 2) that many of the fundamental molecule of life possess conical intersections for rapid radiationless dissipation of the photon-induced electronic excitation energy to the ground state [Michaelian (2017, 2021)], 3) that efficient photochemical routes to production of the fundamental molecules from simple and common precursors, such as HCN and $CO_2$ in water, under UVC light have been found, and that these routes have the hallmarks of dissipative structuring [Michaelian (2017); Michaelian and Rodriguez (2019); Michaelian (2021)], 4) that we have found a DNA and RNA enzyme-less denaturing mechanism tied to UVC photon dissipation [Michaelian and Santillán Padilla (2014); Michaelian and Santillan (2019)], 5) that the homochirality of life can be explained from the morning/afternoon temperature and UVC photon circular polarization asymmetry at the ocean surface and the temperature dependence of UVC-induced denaturing [Michaelian (2018)], 6) that the strong chemical affinity of the UVC absorbing amino acids (the aromatics), and others, to their codons and anticodons can be explained based on the thermodynamic selection of greater photon dissipation afforded to the complex [Mejía Morales and Michaelian (2020)], 7) that dissipative structuring of the fundamental molecules under UVC light provides a simple explanation for their existence in space and on other astronomical bodies [Michaelian and Simeonov (2017)], and, 8) that plants appear to optimize evapotranspiration (the water cycle) over photosynthesis [see Michaelian (2012a, b) and references therein].

We welcome and appreciate all challenges to our Thermodynamic Dissipation Theory for the Origin and Evolution of Life.

*Author contributions.* K. Michaelian and A. Simeonov contributed to this Reply.

*Competing interests.* The authors declare no competing interests.

**References**

Barry, R. and Chorley, R.: Atmosphere, Weather, and Climate, 6th edn., 1992.

Coakley, J.: REFLECTANCE AND ALBEDO, SURFACE, in: Encyclopedia of Atmospheric Sciences, edited by Holton, J. R., pp. 1914–1923, Academic Press, Oxford, https://doi.org/https://doi.org/10.1016/B0-12-227090-8/00069-5, 2003.

Couradeau, E., Karaoz, U., Lim, H., and et al.: Bacteria increase arid-land soil surface temperature through the production of sunscreens, Nat Commun, 7, 10 373, https://doi.org/10.1038/ncomms10373, 2016.

Gates, D. M.: Biophysical Ecology, Springer-Verlag, 1980.

Kleidon, A.: Entropy Production by Evapotranspiration and its Geographic Variation, Soil & Water Res., 3, S89–S94, 2008.

Kleidon, A., Fraedrich, K., and Heimann, M. A.: Green Planet Versus a Desert World: Estimating the Maximum Effect of Vegetation on the Land Surface Climate, Climatic Change, 44, 471–493, https://doi.org/10.1023/A:1005559518889, 2000.

Makarieva, A. M. and Gorshkov, V. G.: Biotic pump of atmospheric moisture as driver of the hydrological cycle on land, Hydrology and Earth System Sciences, 11, 1013–1033, https://doi.org/10.5194/hess-11-1013-2007, 2007.

Mejía Morales, J. and Michaelian, K.: Photon Dissipation as the Origin of Information Encoding in RNA and DNA, Entropy, 22, https://doi.org/10.3390/e22090940, 2020.

Michaelian, K.: Thermodynamic stability of ecosystems, Journal of Theoretical Biology, 237, 323 – 335, https://doi.org/https://doi.org/10.1016/j.jtbi.2005.04.019, 2005.

Michaelian, K.: Thermodynamic origin of life, ArXiv, https://doi.org/10.5194/esd-2-37-2011, 2009.

Michaelian, K.: Thermodynamic dissipation theory for the origin of life, Earth Syst. Dynam., 224, 37–51, 2011.

Michaelian, K.: The Biosphere, chap. The biosphere: A thermodynamic imperative, INTECH, 2012a.

Michaelian, K.: Biological catalysis of the hydrological cycle: lifes thermodynamic function, Hydrol. Earth Syst. Sci., 16, 2629–2645, https://doi.org/10.5194/hess-16-2629-2012, 2012b.

Michaelian, K.: Thermodynamic Dissipation Theory of the Origina and Evolution of Life: Salient characteristics of RNA and DNA and other fundamental molecules suggest an origin of life driven by UV-C light, Self-published. Printed by CreateSpace. Mexico City. ISBN:9781541317482., 2016.

Michaelian, K.: Microscopic Dissipative Structuring and Proliferation at the Origin of Life, Heliyon, 3, e00 424, https://doi.org/10.1016/j.heliyon.2017.e00424, 2017.

Michaelian, K.: Homochirality through Photon-Induced Denaturing of RNA/DNA at the Origin of Life, Life, 8, https://doi.org/10.3390/life8020021, 2018.

Michaelian, K.: The Dissipative Photochemical Origin of Life: UVC Abiogenesis of Adenine, Entropy, 23, https://doi.org/10.3390/e23020217, 2021.

Michaelian, K. and Rodriguez, O.: Prebiotic fatty acid vesicles through photochemical dissipative structuring, Revista Cubana de Química, 31, 354–370, 2019.

Michaelian, K. and Santillan, N.: UVC photon-induced denaturing of DNA: A possible dissipative route to Archean enzyme-less replication, Heliyon, 5, e01 902, https://www.heliyon.com/article/e01902, 2019.

Michaelian, K. and Santillán Padilla, N.: DNA Denaturing through Photon Dissipation: A Possible Route to Archean Non-enzymatic Replication, bioRxiv, https://doi.org/10.1101/009126, 2014.

Michaelian, K. and Simeonov, A.: Fundamental molecules of life are pigments which arose and co-evolved as a response to the thermodynamic imperative of dissipating the prevailing solar spectrum, Biogeosciences, 12, 4913–4937, 2015.

Michaelian, K. and Simeonov, A.: Thermodynamic explanation of the cosmic ubiquity of organic pigments, Astrobiol. Outreach, 5, 156, 2017.

225 Muhsin, M., Sunilkumar, S. V., Venkat Ratnam, M., Krishna Murthy, B. V., and Parameswaran, K.: Seasonal and Diurnal Variations of Tropical Tropopause Layer (TTL) Over the Indian Peninsula, Journal of Geophysical Research: Atmospheres, 122, 12,672–12,687, https://doi.org/https://doi.org/10.1002/2017JD027056, 2017.

Prigogine, I.: Introduction to Thermodynamics Of Irreversible Processes, John Wiley & Sons, third edn., 1967.

Schneider, E. D. and Kay, J. J.: Complexity and thermodynamics: towards a new ecology, Futures, 24, 626–647, 1994.

230 Ustin, S. L., Valko, P. G., Kefauver, S. C., Santos, M. J., Zimpfer, J. F., and Smith, S. D.: Remote sensing of biological soil crust under simulated climate change manipulations in the Mojave Desert, Remote Sensing of Environment, 113, 317–328, https://doi.org/https://doi.org/10.1016/j.rse.2008.09.013, 2009.

Varotsos, C. A., Melnikova, I. N., Cracknell, A. P., Tzanis, C., and Vasilyev, A. V.: New spectral functions of the near-ground albedo derived from aircraft diffraction spectrometer observations, Atmospheric Chemistry & Physics, 14, 6953–6965, https://doi.org/10.5194/acp-14-
235 6953-2014, 2014.

Wang, J., Bastiaanssen, W. G. M., Ma, Y., and Pelgrum, H.: Aggregation of land surface parameters in the oasis–desert systems of north-west China, Hydrological Processes, 12, 2133–2147, https://doi.org/https://doi.org/10.1002/(SICI)1099-1085(19981030)12:13/14<2133::AID-HYP725>3.0.CO;2-6, 1998.

---

## Author Comment (AC3)

Reply to RC2: 'Comment on bg-2021-135', Axel Kleidon, 15 Jul 2021

I would like to first thank both the original authors and the reviewers for interesting discussion and for directing my attention to interesting literature that I did not know of before.

I used the Moon for comparison, because I had no Earth without life at hand with which to compare the present vegetated Earth. Life probably did not originate on land, but I shall return to the situation in water. The best proxy for life on land may be what Zang et al. (2013) have described. They compared the albedo of nonpopulated desert sand with that having a biological crust consisting of cyanobacteria, green algae, mosses, lichens, and other organisms. They found that only if the surface was very dry was the albedo higher for the unpopulated sand Fig, 1).

[Figure]

Fig. 1. Albedo for desert dune sand with and without biological soil crust organisms (BSC) at different moisture contents. From Zang et al. (2013).

Kleidon et al. (2000) deal primarily with the land surface. Many researchers believe that life originated in an aquatic environment. In fact, there probably was not much land when life originated; see, e.g., Liu et al. (2021). Kleidon writes (RC2): "Furthermore, it is textbook knowledge that the surface albedo of forests is in most cases darker than bare ground, even though there may be some isolated exceptions." Again, it is probably not the land surface, and certainly not forests, that are most relevant in a discussion about the origin and early evolution of life. To the situation for early life, the conditions in an aquatic environment are more relevant. For a long time life and photosynthesis probably took place mostly underwater.

For the ocean, algal growth decreases the reflectance at shorter wavelengths, increases it at longer wavelengths (Fig. 2, from Bacour et al. 2020).

[Figure]

Fig. 2. Ocean column reflectance for eight values of the chlorophyll concentration (0.03, 0.1, 0.3, 0.5, 1, 3, 5, 10 mg.m$^{-3}$) computed by the Coupled Ocean and Atmosphere Radiative Transfer (COART) bio-optical model (https://cloudsgate2.larc.nasa.gov/jin/coart.html, accessed on 15 March 2020; see also https://cloudsgate2.larc.nasa.gov/jin/rtnote.html). From Bacour et al. (2020). It should be noted that spectral reflectance graphs published by Li et al. (2021), although similar, differ quantitatively from these, with a single peak for 10 mg m$^{-3}$ chlorophyll $a$. Note that the values shown in Fig. 6 of Li et al. (2021) should be multiplied by $2\pi$ (conversion from reflectance over 1 radian to reflectance over $2\pi$ radians) to be comparable with the values in this Figure.

Similar results are given by Li et al. (2021) (Fig. 3), although the spectral and angular coverage is not quite sufficient for reliable computations of solar energy reflectance or entropy changes. Algae in the ocean decrease reflectance below 510 nm, increase reflectance at wavelength longer than 510 nm. Higher reflectance in this case means lower entropy production. These reflectance spectra should be compared to Archaean daylight (Fig. 5). It is not quite certain whether reflectance increases or decreases with increasing amount of photosynthetic plankton. This partly depends on what early photosynthetic pigment we assume. The integrated reflected radiant powers for 0.03 mg chlorophyll per m$^3$ of Bacour et al. (2020), i.e. 2.66 W m$^{-2}$, and Li et al. (2019), 2.77 W m$^{-2}$, do not differ significantly from each other. In the values for 10 mg chlorophyll per m$^3$, however, the two publications differ. Bacour et al. (2020) give 2,87 W m$^{-2}$, a somewhat higher value than for 0.03 mg chlorophyll per m$^3$, and Li et al. (2019) 2,18 W m$^{-2}$, i.e. lower than for 0.03 mg chlorophyll per m$^3$. Using the reflectances of Jin et al. (2011) the values are 1.02 W m$^{-2}$ for 0 mg chlorophyll per m$^3$ and 1.32 W m$^{-2}$ for 10 mg chlorophyll per m$^3$, i.e. more reflected power with phytoplankton. Also these spectra are a bit truncated, and it is likely that the difference would be larger with wider spectral coverage. In conclusion it can be said

that at present we cannot conclude whether the presence of phototrophs in the ocean in general increased or decreased reflected power from the sun during the Archaean.

[Figure]

Fig. 3. Ocean reflectance over 1 radian according to Li et al. (2021) for various concentrations of chlorophyll *a*-containing organisms.

[Figure]

Fig 4. Ocean reflectance without and with phytoplankton with chlorophyll according to Jin et al. (2011).

[Figure]

Fig. 5. Daylight spectrum at ground level in the Archaean according to various estimates compiled by Arney et al. (2016). I have used the data marked "Khare-Hasenkopf" (red curve) for my calculations of reflected specteal power in the Fig. 5.

[Figure]

Fig. 6. The convolution of the red curve in Fig. 4 by the spectra in Figs 2 and 3 for 0,03 and 10 mg chlorophyll per m³. The spectra of Li et al. in Fig. 3 were multiplied by 2π to for adjustment to same angular geometry.

[Figure]

Fig, 7. he convolution of the red curve in Fig. 4 by the spectra in Fig. 4.

It must be assumed that the power that is not reflected, ultimately will give rise to radiation of longer wavelength, mostly heat radiation of the temperature of the Earth surface. I thought at first that for life in water we do not have to consider the liquid to gaseous transition of water when discussing entropy and early life. This is, however, not correct. The presence of pigmented organisms can cause local heating of up to 1.5°C of the ocean surface, and thus probably cause extra evaporation (Kahru et al. 1993).

Michaelian and Simeonov say (bg-2021-135-CC4-supplement, lines 103–104): "Less than 0.1% of the free energy in sunlight goes into carbon bond making, which is how photosynthesis stores free energy [Gates (1980)]." The same is also said by Simeonov in the earlier bgd-12-C1904-2015, also referring to Gates (1980). This may be correct, although I have not been able to find support for it in Gates (1980), who writes (p. 491) "Of the total amount of light energy incident on a plant, only 1 to 7% is converted to useful products". Zhu et al. (2010) specify "an efficiency of conversion of full-spectrum solar radiation into biomass of approximately 1.5%" for soybean (p. 240), and "The theoretical maximal photosynthetic energy conversion efficiency (εc) is 4.6% for C3 and 6% for C4 plants" (p. 240), although this would perhaps never be realized.

References

Arney, G., Domagal-Goldman, S.-D., Meadows, V.S., Wolf, E.T., Schwieterman, E., Charnay, B., Claire, M., Hébrard, E. & Trainer, M.G. (2016) The pale orange dot: The spectrum and habitability of hazy Archean Earth. Astrobiol. 16, 873–899. DOI: 10.1089/ast.2015.1422

Bacour, C., Bréon, F.-M., Gonzalez, L., Price, I., Muller. J.-P. & Straume, A.G. (2020) Simulating Multi-Directional Narrowband Reflectance of the Earth's Surface Using ADAM (2020) A Surface

Reflectance Database for ESA's Earth Observation Missions. Remote Sens. 12: 1679 (24 pp.). doi:10.3390/rs12101679

Jin, Z., Qiao, Y., Wang, Y., Fang, Y. & Yi, W. (2011) A new parameterization of spectral and broadband ocean surface albedo. Optics Express 19, 26429–26443.

Kahru, M., Leppanen, J.M. & Rud, O. (1993) Cyanobacterial blooms cause heating of the sea surface. Mar. Ecol. Prog. Ser. 101, 1–7.

Kleidon, A., Fraedrich, K. & Heimann, M. (2000) A green planet versus a desert world: estimating the maximum effect of vegetation on the land surface climate. Climatic Change 44, 471–493.

Li, J., Li, T., Qingjun Song, Q. & Ma, C. (2021) Performance evaluation of four ocean reflectance model. Remote Sens. 13: 2748 (17 pp.).

Liu, C.-T. & He, Y.-S. (2021) Rise of major subaerial landmasses about 3.0 to 2.7 billion years ago. Geochem. Persp. Let. 18, 1–5 | doi: 10.7185/geochemlet.2115

Xiao, B. & Bowker, M.A. (2020) Moss-biocrusts strongly decrease soil surface albedo, altering land-surface energy balance in a dryland ecosystem. Sci. Total Environm. 741: 140425 (15 pp.). https://doi.org/10.1016/j.scitotenv.2020.140425

Zhang, Y.-f., Wang, X.-p., Hu, R., Pan, Y.-x. & Zhang, H. (2013) Variation of albedo to soil moisture for sand dunes and biological soil crusts in arid desert ecosystems. Environ. Earth Sci. 71, 1281–1288. DOI 10.1007/s12665-013-2532-7

Zhu,X.-G., Long, S.P. & Ort, D.R. (2010) Improving photosynthetic efficiency for greater yield. Annu. Rev. Plant Biol. 61:235–261.

---

## Author Response (AR1)

**Response by Lars Olof Björn to bg-2021-135-CC4-supplement**

This is a revision of an earlier manuscript.
In the new manuscript new sections are shaded in grey

Major changes from previous manuscript:
The following references have been deleted:
Foote Smith, E, (2019), Meireles, J.E. et al. (2020), Rautiainen, M. et al, (2018)
The following of the former Figures have been deleted: Figure 1, Figure 2, Figure 3, Figure 5
The following Figures are new: Figure 2, Figure 3, Figure 5, Figure 6
Figure 4 was modified (correction of comma to dot, change from color to grey tone).

One of the reviewers had no complaints about the manuscript. Here follow remarks by the other reviewer (in blue) and my response (in black).

**Kleidon**

1. Is the claim by Michaelian & Simeonov (2015) valid or useful? Here I side with the skeptical attitude of the author of the commentary and with William Martin. I do not believe MS2015, and doubt that it has much value. So there would be quite a bit that can be criticized, but this criticism in its own needs to be well justified and substantiated.
2. Is the commentary by Björn well justified and substantiated? As I expressed in my review, it is not. For a commentary, I would expect something stronger and better argued, and I gave an example of how such a commentary could look like in the review. In the case of MS2015, I think that such a more substantial commentary would be easy to do.

Specifically, the comment questions the statement that "Living systems reduce the albedo of Earth" and elaborates whether this statement is correct. The manuscript uses the example of the Moon and compares it to the Earth, and provides some anecdotal evidence where life is not darker than its surroundings. Yet, this comparison is flawed, because what we would need to compare is an Earth without life to an Earth with life, not with another planetary body (Simulations of such conditions have been made, e.g., Kleidon et al. (2000) "A green planet versus a desert world", Clim Change, 44: 471-493.). Furthermore, it is textbook knowledge that the surface albedo of forests is in most cases darker than bare ground, even though there may be some isolated exceptions.

**Björn reply**: I have cut down on comparison with the Moon, added some remarks about Mars, but extended the article with comparisons between areas on Earth with and without organisms, and one comparison that shows that a type of prokaryote (cyanobacteria) reflects more sunlight that eukaryotes in the same habitat. For terrestrial (land) areas I have focused on bacteria and biocrusts, and deleted the parts about trees, since we should discuss the early stages of biological evolution. Therefore I have also added more information about aquatic life, since I believe that life arose in water.

Also the original authors criticized my comments to their article. Their main views are shown below (in blue) followed by my response (in black).

**Michaelian & Simeonov**

Michaelian and Simeonov (2015a) do not "call everything that absorbs photons a pigment". (See point 4 of the comment by Michaelian and Simeonov (2015b) on our original article for validation of the use of the word "pigment" in our paper.) We suggest that those organic molecules now known as the fundamental molecules of life (i.e. those in the 3 domains of life) that strongly absorb light within the 210-285 nm (UVC) region and have a conical intersection to rapidly dissipate the electronic excitation energy into heat (Michaelian2011;2017;2021) were originally (at the origin of life) organic pigments which were dissipatively structured from simpler and more common precursor molecules under this UVC light to perform the thermodynamic function of dissipating this light into heat.

Björn bases his critique on the suggestion that non-living material can be more photon absorbing than living material. He gives a number of examples in which he shows that the albedo of material devoid of life is lower than that of 5 biotic material and concludes that these examples counter our assertion. However, Björn makes the erroneous assumption that albedo (reflection) is the only important factor related to photon dissipation (entropy production) occurring in the light-pigment interaction in living systems. He ignores the other contributions to entropy production due to the photon interaction which were listed in our article;
1) the shift towards the infrared of the emitted spectrum,
2) the diffuse emission and reflection of light into a greater outgoing solid angle,
3) the coupling of photon-induced evapotranspiration in the pigmented leaf to further photon dissipating processes such as the water cycle, which further allows dissipating biopigments to flourish over all of Earth's surface.

**Björn reply**:
1)       I try to make more clear that I include the shift towards the infrared of the emitted spectrum. The more of the incident radiation that is absorbed, the more is eventually emitted as infrared radiation (p. 6, lines 111–113).
2)       I do not believe that there is any difference in principle between living and dead matter as to the angular distribution of reflected radiation (see p. 1, lines 20–22). Except for smooth wet surfaces and calm water surfaces, most natural surfaces reflect light in a diffuse way.
3)       Yes, this is a point, in particular as regards life on land. Therefore I have deleted leaves and put more emphasis on aquatic life, which is also more relevant, because the earliest life is thought to have been aquatic. Also aquatic life can in some cases have an effect on evaporation, as can life in the atmosphere have an effect on precipitation by serving as condensation nuclei. Thus Mati Kahru, Juha-Markku Leppänen & Ove Rud (Cyanobacterial blooms cause heating of the sea surface, Marine Ecol. Progr. Ser. 101, 1–7, 1993) noted that cyanobacteria can cause increased surface temperature, and presumably increased

evaporation, from the water surface. I have included a reference to this work, but have assumed that this is a minor and not very common effect. I have now mentioned this on p. 4, lines 82–84.

Michaelian & Simeonov also have more detailed criticism in a supplement (bg-2021-135-CC4-supplement.pdf). Here follows my response to that:

**Reply to bg-2021-135-CC4-supplement by Karo Michaelian and Aleksandar Simeo**nov

**Abstract** (of KM's and AS's supplement)
KM and AS write (lines 5–6) "Björn makes the erroneous assumption that albedo (reflection) is the only important factor related to photon dissipation (entropy production) occurring in the light-pigment interaction in living systems." This is a misunderstanding. I fully understand that the change in angular distribution and change of the radiation to longer waves are also important.

Forests (e.g., Rengarajan & Schott (2017) as well as moss and lichen (e.g., Solheim et al. 2000) exhibit large deviations from Lambertian scattering, but so do some, but not all, unvegetated natural ground surfaces (Watson 1972, Pommerol et al. 2013, Hapke 2021). The general tendency seems to be that vegetated surfaces have more reflection concentrated in the direction opposing the incident light, than do unvegetated surfaces. This would contribute to less entropy generation by vegetated surfaces.
Literature cited above:
**Hapke**, B. (2021) Bidirectional reflectance spectroscopy 8. The angular width of the opposition effect in regolith-like media. Icarus 354: 114105 (13 pp.). https://doi.org/10.1016/j.icarus.2020.114105
**Rengarajan**, R. & **Schott**, J.R. (2017) Modeling and simulation of deciduous forest canopy and its anisotropic reflectance properties using the Digital Image and Remote Sensing Image Generation (DIRSIG) tool. IEEE J. Selected Topics Appl. Earth Observ. Remote Sens. (14 pp.).DOI: 10.1109/JSTARS.2017.2751539
**Pommerol**, A., Thomas, N., Jost, B., Beck, P., Okubo, C. & McEwen, A.S. (2013) Photometric properties of Mars soils analogs. Geophys. Res. Planets 118, 2045–2072, doi:10.1002/jgre.20158
**Solheim**, I., Engelsen, O., Hosgood, B. & Andreoli, G. (2000) Measurement and modeling of the spectral and directional reflection properties of lichen and moss canopies. Remote Sens. Environ. 72, 78–94.
**Watson**, R.D. (1972) Spectral Reflectance and Photometric Properties of Selected Rocks. Remote Sens. Environ. 2, 95–100.

The heat radiation emitted (and fluorescence, usually a minor part of the emitted radiation) over wider angles than the incoming light depends on the amount of radiation absorbed, and thus on albedo (although for fluorescence also other circumstances are important). KM and AS further write: "The coupling of photon-induced evapotranspiration in the pigmented leaf to further photon dissipating processes such as the water cycle, which further allows dissipating biopigments to flourish over all of Earth's surface." It is true that phase transitions can also be important, but since the title of the original publication is "Fundamental molecules of life are **pigments which arose** and co-evolved as a response to the thermodynamic imperative of dissipating the prevailing solar spectrum" we should not be concerned with leaves and land plants, since no new photosynthetic pigments have evolved after the transition of life from the aquatic environment to land. I have removed reference to land plants from the revised manuscript. My intended main emphasis is on aquatic organisms, since life with all probability started in an aquatic environment. As for life on land, I concentrate on "simple" organisms (if

we can rightly label any organisms as simple). My Figure 3 is intended to demonstrate that cyanobacteria have higher albedo than eukaryotes inhabiting the same environment. This could, with good will, be interpreted as if organisms evolve toward increased absorption.

KM and AS further criticize my sentence: "Thus, it appears that if Michaelian and Simeonov are correct, one would expect organisms, in particular phototrophic organisms, or the biosphere to be less reflecting and more absorbing than dead matter." I have removed this offending sentence from the new version.

**Albedo of the Moon, a world without life**
On line 66 KM and AS write "a wavelength dependent albedo will not change the result". This is surprising. I thought that high albedo at large wavelength would result in more entropy production. Low albedo at low wavelengths will, in principle result in as much thermal radiation per incident flux as low albedo at large wavelengths, while long-wave reflected light would carry more entropy than short-wave reflected light. KM and AS write on lines 88–89 "it is also important to consider the temperature data of the emitted spectrum".

**Comparison of planets**
On lines 40–42 KM and AS write: "We found that Earth's entropy production per unit area is almost twice that of either of its neighbors, and this may be attributed to the presence of life on Earth [Michaelian (2012b)]." I think that it rather may be related to the different influx of solar energy, i.e. to the different distances from the Sun. If we adjust their values for this, we find (using the values in their Table 1 and leaving out the units):
For the Earth $1.247 \times (1.4960 \times 10^{11})^2 = 2.791 \times 10^{22}$
and for Mars $0.689 \times (2.2792 \times 10^{11})^2 = 3.619 \times 10^{22}$, i.e. a ca 30% higher value for Mars, despite the fact that the Earth has oceans and a thicker atmosphere.

**The temporal aspect**
KM and AS write (line 101): "The forests, as Björn correctly indicates, are sometimes buried and later burned as fossil fuel by humans. However, they produced at least 1000 times more entropy during their lifetime than that obtained by burning the same trees as fossil fuel today." I do not see that they have got my point. A tree lives, say 100 years. Assuming the "1000 times more entropy during their lifetime than that obtained by burning the same trees as fossil fuel today" is correct, the fraction 0.001 of the entropy not produced during the lifetime is "saved" during 325 million years and emitted now. 325 000000 million years is 325 000 times the life-time of the tree, thus causing a substantial delay (lowered rate) of the production of the 0.001 of the entropy, thus causing a substantial decrease of the entropy production rate.

On line 116 KM and AS correctly point out that "fluorescence reduces entropy production", followed on lines 118–119 "ecosystems still have room to evolve under the thermodynamic imperative towards becoming even better dissipating systems". However, it appears that the fluorescence yield of the photosynthetic system of plants and algae is already optimized for efficient photosynthesis. The chemical potential that can be achieved by a pigment in photosynthesis is a positive function of the fluorescence yield. See, e.g., equation (6) in

"Thermodynamics of light emission and free energy storage in photosynthesis" by Robert T. Ross and Melvin Calvin, Biophys. J. 7, 595–614, 1964. On the other hand, only energy that is not emitted as fluorescence can be used to "split water".

---

## Author Response (AR2)

Response to comments (bg-2021-135, L.O. Björn)

L28-30. Making comparisons to Mars and the moon take away from the narrative. Let the focus purely be on Earth and the evolution of living systems on Earth.
**Deleted.**

Section. 2 Vegetation compared to bare ground.

I think the changing albedo of glaciers/ice sheets due to ice algae should be touched upon in this section.
**Touched upon on lines 174–177**.

L36. I am not sure what point you are trying to make with this lead sentence.
**Sentence deleted.**
L37. You disregard the importance of UV radiation but it could be of importance when thinking about primitive PAH pigments, as well as UV-absorbing pigments in marine algae.
**Now mentioned on lines 68–70.**
L38. Do not start a sentence with 'And'. Also, fix the grammatical errors within this sentence.
**'And' deleted. Hopefully other errors fixed.**
L44. Please clarify. 'ground albedo increases with the proportion of cyanobacteria cover (compared to cover by cyanobacteria, moss, and lichen?
**Sentence deleted.**
L47. Fix, 'De opposite effect'.
**Changed to 'The opposite effect'.**

Section 3 The temporal aspect
L72. Legend. ' Fix. The variation of cyanobacteria coverage was partly natural, partly caused by experimental treatment
**The Figure and its legend have been deleted.**
L76. Fix, 'We cannot back from our responsibility….'
**Deleted.**

Section 4 The aquatic environment.
There are a few extra important points which could be included in this section. These include, the impacts of sea-foam formed from marine algal surfactants and marine aerosols (DMS) and cloud formation.

L80-81. 'since we do not have to deal with gases'. Yes, but we still have to deal with dissolved gases.
Expanded: 'since we do not have to deal with gases (with the exception of dissolved ones, which may escape
**to the atmosphere, and those forming froth).**
L89-90. I would include the sargassum paper by Bach et al. here.
**Added on lines 106–108: ' Bach et al. (2021) suspect that the increased albedo caused by recent increase in the Atlantic Ocean of floating *Sargassum* algae may be more important than the effect of the alga on carbon dioxide and phytoplankton nutrients.'**

L101. A gap is needed. '(2020) and…'.
**Fixed.**

**Further change:** A new Section 2, 'Ancient life', has been added, because the previous manuscript version dealt too little with this.